# CReFT-CAD: Boosting Orthographic Projection Reasoning for CAD via Reinforcement Fine-Tuning

**Ke Niu**[*], **Zhuofan Chen**[*], **Haiyang Yu**[✉], **Yuwen Chen**, **Teng Fu**, **Mengyang Zhao**,
**Bin Li**[✉], **Xiangyang Xue**[✉]

College of Computer Science and Artificial Intelligence, Fudan University, China
`{kniu22,zfchen23,tfu23}@m.fudan.edu.cn`
`{hyyu20,myzhao20,libin,xyxue}@fudan.edu.cn`

## Abstract

Computer-Aided Design (CAD) is pivotal in industrial manufacturing, with orthographic projection reasoning foundational to its entire workflow—encompassing design, manufacturing, and simulation. However, prevailing deep-learning approaches employ standard 3D reconstruction pipelines as an alternative, which often introduce imprecise dimensions and limit the parametric editability required for CAD workflows. Recently, some researchers adopt vision–language models (VLMs), particularly supervised fine-tuning (SFT), to tackle CAD-related challenges. SFT shows promise but often devolves into pattern memorization, resulting in poor out-of-distribution (OOD) performance on complex reasoning tasks. To tackle these limitations, we introduce CReFT-CAD, a two-stage fine-tuning paradigm: first, a curriculum-driven reinforcement learning stage with difficulty-aware rewards to steadily build reasoning abilities; second, supervised post-tuning to refine instruction following and semantic extraction. Complementing this, we release TriView2CAD, the first large-scale, open-source benchmark for orthographic projection reasoning, comprising 200,000 synthetic and 3,000 real-world orthographic projections with precise dimensional annotations and six interoperable data modalities. Benchmarking leading VLMs on orthographic projection reasoning, we show that CReFT-CAD significantly improves reasoning accuracy and OOD generalizability in real-world scenarios, providing valuable insights to advance CAD reasoning research. The code and adopted datasets are available at `https://github.com/KeNiu042/CReFT-CAD`.

## 1 Introduction

Computer-Aided Design (CAD) is now integral to industrial product development, underpinning design, manufacturing, and simulation workflows. In the design phase, engineers use CAD drawings or rasterized orthographic projections for their precision and facile editability. During manufacturing, these drawings are converted into constraint-based parameter tables; for simulation, they yield boundary-representation (B-Rep) data or textual geometry descriptions. A truly user-centric pipeline thus requires accurate semantic parsing of orthographic projections: automated extraction of parameter tables enables direct generation of 3D models meeting stringent manufacturing and simulation standards. By contrast, existing reverse-engineering methods rely on expensive 2D/3D scanning hardware and labor-intensive post-processing, greatly restricting scalability and widespread industrial adoption.

---

[*] Equal contribution. ✉ Corresponding authors.

39th Conference on Neural Information Processing Systems (NeurIPS 2025).

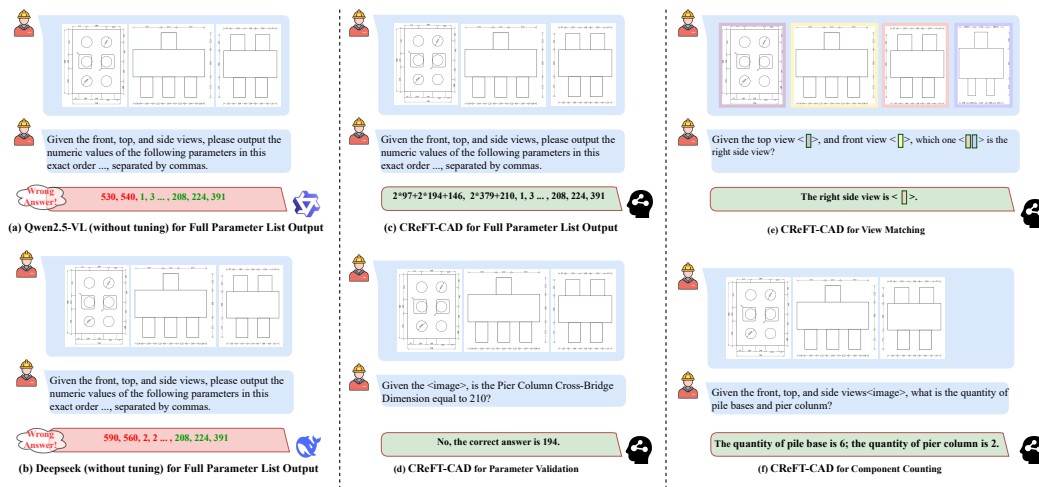

Figure 1: (a) (b) show the results of parameterization tasks of orthographic projection reasoning using Qwen2.5-VL and Deepseek without tuning. (c)–(f) illustrate CReFT-CAD's capabilities across various orthographic projection reasoning tasks.

With the advent of deep learning, orthographic projection reasoning has typically been framed as a standard 3D reconstruction pipeline: models process rasterized drawings to output B-Rep structures [1, 2], point clouds [3, 4], or meshes [5]. However, this paradigm suffers from two critical limitations. First, industrial drawings demand near-zero tolerance for missing or erroneous components: pixel-level discretization errors in reconstruction can propagate to 3D models, leading to failures in downstream manufacturing and simulation. Second, practical CAD workflows require parametric editability and precise semantic alignment—key capabilities that common 3D representations lack. While some methods generate CAD command sequences to recover editability, they still overlook the nuanced semantics of orthographic views, thus failing to reliably map 2D drawing features to their intended 3D counterparts.

Recent works leverage the modality alignment and semantic reasoning capabilities of vision–language models (VLMs) [6–12], addressing CAD challenges via supervised fine-tuning (SFT) [13–15]. However, orthographic projection reasoning requires genuine inferential ability, and task-specific SFT often induces pattern memorization and out-of-distribution (OOD) performance degradation rather than fostering deeper reasoning [16]. Inspired by DeepSeek R1-Zero's Group Relative Policy Optimization (GRPO) [17], we propose Curriculum-driven Reinforcement Fine-tuning for CAD (CReFT-CAD).

CReFT-CAD is a versatile framework enabling interactive orthographic projection reasoning across the full lifecycle of industrial design—from design and manufacturing to simulation (see Fig. 1). It consists of a two-stage paradigm integrating reinforcement learning and supervised tuning: the first stage is **Curriculum-driven Reinforcement Fine-tuning**, where we introduce a difficulty-aware reward mechanism that incrementally increases task complexity, gradually exposing the model to increasingly complex reasoning challenges to promote stable policy optimization; the second stage is **Supervised Post-tuning**, where we build on the reinforced model and further refine its instruction-following and reasoning capabilities via multi-task SFT. Our qualitative and quantitative evaluations show that CReFT-CAD achieves robust orthographic projection reasoning and strong generalization in OOD scenarios. Another key barrier in orthographic projection reasoning is the scarcity of open-source datasets with high-fidelity annotations. To address this, we present TriView2CAD, the first large-scale benchmark tailored to industrial CAD pipelines. Leveraging real-world design archives, we use a constraint-driven synthesis pipeline to generate 200,000 synthetic and 3,000 real-world three-view sets, each annotated with precise, one-to-one dimensional labels linked to their corresponding geometric primitives. This design allows models to extract quantitative measurements from rasterized drawings and enforce exact geometric constraints for 3D reconstructions ready for simulation.

## 2 Related Work

### 2.1 Orthographic projection reasoning for CAD

Understanding and reasoning with orthographic projection is a fundamental and longstanding challenge in CAD-related tasks [18–20]. With the development of 3D reconstruction methods, existing methods for orthographic projection reasoning have largely centered on standard 3D reconstruction pipelines. SPARE3D [21] introduces a dataset for evaluating the spatial reasoning capabilities of AI systems via 2D line drawings of 3D objects. Contrastive-SPARE3D [22] proposes a self-supervised binary classification network that helps learn 3D object line drawing representations that are detail-sensitive and view-invariant. PlankAssembly [23] presents a transformer-based sequence generation model that learns flexible input-output mappings. IsoTGAN [24] introduces a novel Gaussian-enhanced Euclidean attention mechanism and a geometric constraint loss function to further enhance local image features. GaussianCAD [25] employs a custom sparse-view 3D reconstruction method, removing reliance on vector CAD sketches and real-world 3D data.

### 2.2 Applications of LVLMs in CAD-Related tasks

Recent efforts applying vision–language models (VLMs) to CAD tasks fall into three main categories:**1) Direct Adaptation of Pretrained VLMs.** CAD-Recode [26] leverages a VLM to translate raw point cloud inputs into executable Python code.Both LEAM [27] and LLM4CAD [28] generate CAD models by jointly leveraging text descriptions and image inputs. CAD-Assistant [29] introduces a novel approach using VLMs as planners, where the model interacts with Python APIs to accomplish diverse CAD tasks. **2) Supervised Fine-Tuning (SFT).**CAD2Program [13] generates 3D parametric models from 2D CAD drawings. CAD-MLLM [30] enables parametric CAD modeling from text, images, and point clouds, and introduces the Omni-CAD dataset. **3) Reinforcement Learning–based Tunings.**RLCAD [31] introduces a reinforcement learning environment for generating complex CAD models. CADCrafter [32] presents an image-to-CAD generation framework, fine-tuned via Direct Preference Optimization (DPO) to directly translate input images into executable CAD representations.

## 3 TriView2CAD: Benchmarking orthographic projection reasoning

As previously noted, a critical barrier in CAD research is the absence of an open-source dataset with high-fidelity annotations for orthographic projection reasoning. Existing benchmarks focus on 3D reconstruction from rasterized 2D views, yet diverge from real-world engineering drawing interpretation in three key aspects:

**Lack of precise dimensional annotations.** Existing orthographic projection benchmarks provide only rasterized drawings without precise dimension annotations. This causes models to prioritize relative scale over exact measurements, and pixel-level errors in rasterized images can easily propagate into the reconstructed 3D model, resulting in inaccuracies. Additionally, the limited editability of reconstructed 3D models prevents corrective adjustments during design and simulation.

**Lack of explicit logical reasoning tasks.** Existing datasets do not evaluate a model's inferential capabilities within or across orthographic views. Consequently, models trained on these benchmarks learn pixel-level reconstruction skills but fail to handle real-world challenges such as inferring omitted annotations or leveraging structural symmetries.

**Lack of essential CAD modalities.** Most datasets contain only one or two modalities—typically raster images and, at best, a single 3D format (meshes or point clouds). This limited coverage omits critical data representations (e.g., parameter tables, vector CAD files, executable commands, STEP/B-Rep) required for end-to-end CAD design, manufacturing, and simulation pipelines.

### 3.1 Dataset construction

In this work, we focus on prefabricated bridge piers owing to their inherently modular structure—composed of repeated, standardized components—that enables a compact yet expressive parameter space for dataset synthesis. Moreover, prefabricated piers are ubiquitous in modern infrastructure and backed by extensive CAD drawing and manufacturing documentation archives.

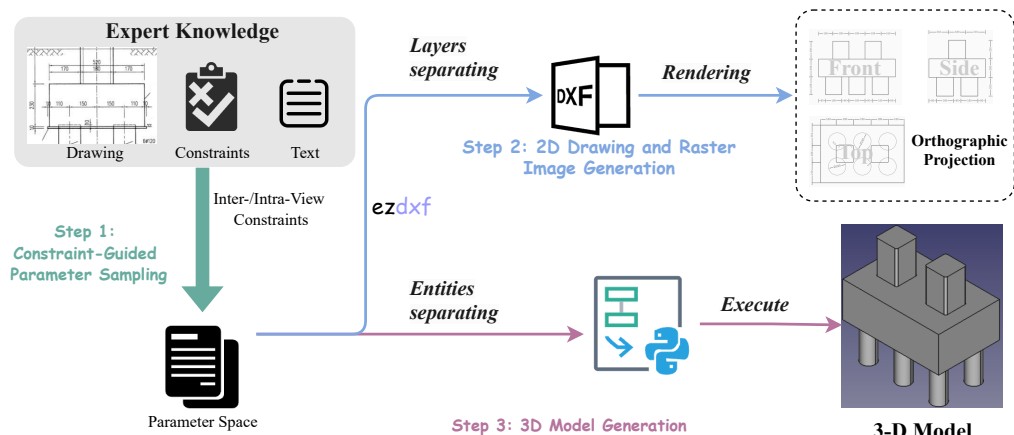

Figure 2: Constraint-driven synthesis pipeline for TriView2CAD.

TriView2CAD comprises 200,000 synthetic samples and 3,000 real-world samples. The synthetic data are split 80/20 into training and test sets; due to the scarcity of real-world engineering drawings, all 3,000 real-world samples are reserved exclusively for testing. The synthetic test set evaluates in-domain performance. As shown in Fig. 3, real-world images typically feature more complex structures—primarily due to redundant annotation lines (causing occlusions and overlaps with other components) and the inclusion of non-numeric annotations. These samples thus assess out-of-distribution (OOD) generalization in authentic CAD scenarios. To align with end-to-end industrial workflows, TriView2CAD provides six interconnected data modalities per sample: structured parameter tables (JSON), vector CAD drawings (DXF), raster images (PNG), executable modeling scripts, and two standard 3D formats (STEP and B-Rep). This comprehensive modality suite supports tasks spanning early design to downstream manufacturing and simulation. As shown in Fig. 5, we adopt a constraint-driven synthesis pipeline, with detailed steps outlined below:

**Step 1: Constraint-Guided Parameter Sampling** We first analyze real-world CAD drawings to define a 15-dimensional parameter space, governed by two constraint classes. **Intra-View Constraints** enforce topological closure and physical validity within each projection: every component must form gap- and overlap-free contours, and paired dimensions adhere to domain-specific engineering constraints (e.g., "Cross-Bridge Pier Spacing" < "Cap Beam Cross-Bridge Dimension"). **Inter-View Constraints** ensure cross-projection consistency by requiring measurements annotated in one view to exactly match their counterparts (height, width, depth) in other views. By embedding these constraints into the sampling pipeline, we ensure all synthetic designs are geometrically coherent and compliant with real-world engineering standards.

**Step 2: 2D Drawing and Raster Image Generation** We use the ezdxf library to convert each sampled parameter vector into a vectorized 2D CAD drawing (DXF format), mapping numeric dimensions to geometric primitives (e.g., lines, circles, arcs) to form coherent orthographic projections. We utilize ezdxf's layer separation functionality to partition primitives into semantic layers, enhancing the editability of the resulting CAD drawings. Each DXF file is then imported into FreeCAD [33], where high-resolution screenshots are captured to generate three orthographic views (front, top, side).

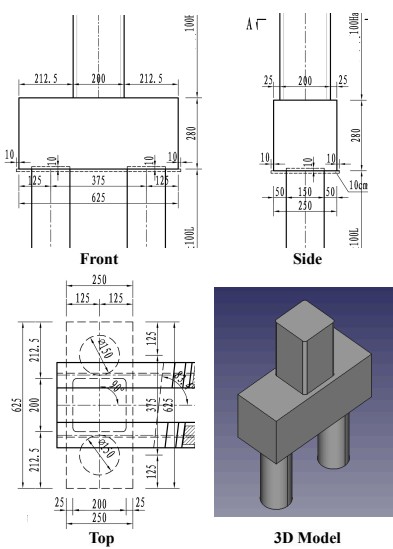

Figure 3: Examples of Real-World Orthographic Projections and its 3D model.

**Step 3: 3D Model Generation** We utilize the FreeCAD Python API to reconstruct a 3D model from its corresponding parameter vector. Consistent with our 2D generation pipeline, we decompose the

| Training Tasks | Input Prompt | Response |
|---|---|---|
| Dichotomous choice task | *Given the front, top, and side views, please refer to the required field names below, and determine whether the given parameter values ...match the image.* | Yes/NO |
| Multiple choice task | *Given the front, top, and side views, please refer to the required field names below, and choose all options that match the image. ...Options: A. 860, 833, ..., 356 B. 860, <mask>, .., 356  C... D ...* | A/B/C/D |
| Parameterization task based on CoT | *Given the front, top, and side views, please output only the numeric values of the following parameters in this exact order..., separated by commas.* | 2*82+2*188+148, 2*398+160, ..., 222 |

Figure 4: Diagram design of three tasks in curriculum-driven reinforcement fine-tuning.

model into discrete entities and generate scripted commands to precisely position each primitive in compliance with the intra- and inter-view constraints defined in Step 1. Executing these scripts automatically produces industry-standard STEP and B-Rep files, closing the loop from parameter sampling to editable, simulation-ready CAD geometries.

## 3.2 Evaluation of TriView2CAD

We benchmark seven leading vision–language models on TriView2CAD to assess their high-precision orthographic projection reasoning capabilities. The evaluation results are presented in Sec. 3.2. Aligning with real-world CAD workflows, we design three complementary evaluation tasks to rigorously test a model's ability to extract precise dimension annotations from rasterized orthographic projections and infer the underlying geometric relationships: **(i) Dimension recognition and pairing** Models identify each annotated dimension in a single orthographic view and map it to its associated geometric feature. **(ii) Primitive counting** Models count instances of specified CAD elements (e.g., number of pier columns), assessing their ability to parse structural composition. **(iii) Composite Parameter Computation** Models compute engineering-critical derived quantities—for example, Cross-Bridge Pier Spacing is calculated as the sum of Pier Column Cross-Bridge Dimension and Pile Spacing. Collectively, we define a 15-dimensional parameter space consisting of 6 recognition parameters, 3 counting parameters, and 6 composite calculation parameters. In evaluation, each parameter is treated as an independent prediction target. Overall accuracy is computed as the total number of correctly predicted parameters divided by the total number of parameters across all test samples. This evaluation suite reveals the strengths and weaknesses of each model in addressing the tightly coupled semantic and quantitative requirements of orthographic projection CAD reasoning.

## 4 Methodology

Inspired by the success of reinforcement learning–based tuning methods [34, 35], we present Curriculum-driven Reinforcement Fine-tuning for CAD (CReFT-CAD), an orthographic-projection reasoning framework. CReFT-CAD integrates a ViT-based visual encoder with Qwen2.5 [36]. First, the framework adopts a curriculum of three progressively challenging reasoning tasks paired with a difficulty-aware reward mechanism, empowering the model to transcend rote pattern matching and develop robust reasoning capabilities while enhancing out-of-distribution (OOD) generalization. Second, Supervised Post-tuning refines the model's instruction-following abilities to accommodate interactive, real-world CAD queries.

### 4.1 Curriculum-driven reinforcement fine-tuning

Orthographic projection reasoning is a sophisticated reasoning task that involves not only dimension recognition and pairing but also primitive counting and more complex composite parameter computation. Building upon GRPO, we propose a Curriculum-driven Reinforcement Fine-tuning (CReFT) strategy, which incrementally exposes the model to increasingly complex training tasks to enhance its reasoning and generalization capabilities. Our task design aligns with the core steps expert engineers follow when verifying CAD drawings, incorporating dichotomous choice tasks, multiple-choice tasks, and chain-of-thought (CoT)-based parameterization tasks. For each task, we design a tailored, efficient reward function to ensure the model can steadily optimize its reasoning capabilities.

**Data engine.** We utilize 160,000 image–text instruction pairs from the synthetic training set of TriView2CAD, where each prompt includes an orthographic projection and all 15 parameter key-value pairs (see Fig. 4).To ensure a balanced training signal across the three tasks, 50% of the training responses are fully correct, while the remaining 50% contain errors of varying degrees. Specifically, for dichotomous choice tasks, each negative response includes $n$ erroneous parameter values, where $n$ follows a uniform distribution $N \sim \text{Uniform}(a, b)$, where $a = 1$ and $b = 15$.

For multiple-choice tasks, $p$ parameter values are masked out in each image-text instruction pair. The masking mechanism is designed to ensure the diversity of distractor options, preventing the model from merely deriving answers through parameter matching—thereby avoiding circumvention of the complex reasoning required for the task. Instead, it compels the model to actively engage in decision-making, evaluating multiple potential solutions and selecting the correct one based on a deeper geometric understanding. For each instruction pair, unmasked values are correct; for incorrect parameter value lists, all unmasked values are correct except for those in $q$ randomly selected erroneous entries. Both the number of masked values $p$ and the number of erroneous entries $q$ follow normal distributions, introducing smooth, gradual variations in task difficulty that enable the model to learn in a more structured, progressive fashion.

For parameterization tasks, we construct image-text instruction pairs grounded in Chain-of-Thought (CoT) reasoning. Specifically, for Composite Parameter Computation tasks, expert knowledge is embedded into step-by-step reasoning prompts to guide the model toward correct outputs. The step-by-step reasoning workflow is as follows: 1) identify the parameters required to compute the composite parameter; 2) specify the calculation formula for the composite parameter; 3) compute the predicted result using the formula and relevant factor parameters.

**Training strategy.** Given that our task involves highly specialized inputs and outputs (as opposed to general-purpose problems), directly applying a pretrained VLM to perform reasoning and generate the required outputs is ineffective. Consequently, the reinforcement learning (RL) reward mechanism cannot function properly—with rewards consistently remaining zero. To mitigate this, we pre-sample a subset of image-text instruction pairs to "warm up" the model before initiating each training task, enabling proper initialization of the RL reward mechanism.

### 4.1.1   Task 1: Dichotomous choice task

For this task, the model's output is restricted to dichotomous responses: "yes" or "no." A "yes" response is returned if and only if all parameters in the 15-dimensional parameter space are fully accurate, whereas any error—even in a single parameter—triggers a "no" response. This stringent criterion incentivizes the model to generate fully accurate parameterizations. Consistent with this design, the reward function is defined as follows:

$$R_{\text{P1}} = \begin{cases} 1, & \text{if all parameters are correct} \\ 0, & \text{otherwise} \end{cases} \tag{1}$$

### 4.1.2   Task 2: Multiple choice task

**Training strategy.** Task 2 introduces multiple-choice tasks to further increase decision-making complexity. These tasks require the model to select from multiple parameter value lists, which significantly enhances the cognitive load compared to binary decisions. Building on the first task, we pre-sample a subset of image-text instruction pairs to conduct instruction-following training on the LLM. This training enables the model to generate structured outputs in the form of multiple-choice questions, facilitating the activation of the reinforcement learning (RL) reward mechanism. Additionally, we design a stringent reward function that encourages the model to fully utilize image-text instruction pairs, ensuring it extracts and leverages all relevant information from both visual and textual modalities. The reward function incentivizes the model to select the most accurate option from the provided choices. Any incorrect selection yields a reward of 0, compelling the model to prioritize precision and reliability in its decision-making.

Let $S_{\text{correct}}$ be the set of correct parameter value lists, $S_{\text{incorrect}}$ be the set of incorrect parameter value lists, and $S_{\text{Selected}}$ be the set of selected parameter value lists by the model. The reward function $R_{p2}$

is defined as:

$$R_{\text{P2}} = \begin{cases} 1 & \text{if } S_{\text{selected}} = S_{\text{correct}} \\ 0.2 & \text{if } S_{\text{selected}} \subseteq S_{\text{correct}} \text{ but } S_{\text{selected}} \neq S_{\text{correct}} \\ 0 & \text{if } (S_{\text{selected}} \cap S_{\text{correct}} \neq \emptyset) \text{ and } (S_{\text{selected}} \cap S_{\text{incorrect}} \neq \emptyset) \\ 0 & \text{if } S_{\text{selected}} = S_{\text{incorrect}} \end{cases} \quad (2)$$

Where $S_{\text{Selected}} = S_{\text{correct}}$ means the model selects only the correct parameter value lists; $S_{\text{Selected}} \supseteq S_{\text{correct}}$ but $S_{\text{Selected}} \neq S_{\text{correct}}$ means the model selects all correct lists but fails to select all of them; $S_{\text{Selected}} \cap S_{\text{correct}} \neq \emptyset$ and $S_{\text{Selected}} \cap S_{\text{incorrect}} \neq \emptyset$ means the model selects both correct and incorrect parameter value lists; $S_{\text{Selected}} = S_{\text{incorrect}}$ means the model selects incorrect parameter value lists.

### 4.1.3 Task 3: Parameterization tasks based on chain-of-thought

**Training strategy.** For this task's training, we start with the model fine-tuned in Task 2 and sample synthetic data to construct standard orthographic projection reasoning QA pairs. These pairs are used for small-batch fine-tuning. After fine-tuning, we assess the difficulty of predicting the 15-dimensional parameters by evaluating the model's performance on a held-out test set. Each parameter's prediction accuracy is categorized into three difficulty levels: **Easy Attributes**: Attributes with an accuracy greater than 0.8. **Medium Attributes**: Attributes with accuracy between 0.2 and 0.8. **Difficult Attributes**: Attributes with accuracy less than 0.2. This difficulty classification is subsequently used to design the reward function. The reward for each correctly predicted attribute is determined by its difficulty level as follows:

$$R_{\text{P3}} = \begin{cases} 1 & \text{if the model correctly predicts an easy attribute} \\ 1.5 & \text{if the model correctly predicts a medium attribute} \\ 2 & \text{if the model correctly predicts a hard attribute} \\ 0 & \text{if the model's prediction is incorrect} \end{cases} \quad (3)$$

This reward function ensures that the model is motivated to gradually improve across all difficulty levels, with greater emphasis on mastering the more challenging tasks, thus enhancing the model's overall reasoning capabilities in orthographic projection tasks.

### 4.2 Supervised post-tuning

Following the CReFT stage—during which the model has developed robust reasoning and generalization capabilities—the focus of Supervised Post-tuning shifts to refining the model's instruction-following capabilities. We revisit the industrial design application context and, adopting a user-centric approach, reformulate tasks in the paradigm of Visual Question Answering (VQA). Leveraging multimodal language models, we design specific tasks that guide the model to output comprehensive parameter lists and make engineering-critical judgments based on orthographic projections. In this stage, the model is trained to handle the following tasks: **1) Full Parameter List Output Task:** Generate a complete and accurate set of parameter key-value pairs— the standard output format for the simulation phase. **2) Parameter Validation Task:** Compare extracted parameter key-value pairs with the ground-truth parameter set in the orthographic projections to verify the accuracy of the extracted parameters. **3) View Matching Task:** Given two distinct orthographic views, determine if the views correspond to the same 3D object or design based on parameter constraints and rasterized images. **4) Component Counting Task:** Identify and count specified components (e.g., bridge piers, columns) in the drawings based on given parameter constraints. This task quantifies design elements to support material estimation and component verification.

Through this post-tuning phase, we strengthen the model's capacity to interpret and respond to complex design queries, thereby enhancing its interactive reasoning capabilities and increasing its applicability to real-world industrial design workflows.

## 5 Experiments

**Implementation Details.** All experiments were conducted on NVIDIA A100 GPUs. Most experiments used GOT-OCR2.0 as the base model, trained on a single server with 8 A100 GPUs and

Table 1: Performance comparison of various VLMs on orthographic projection reasoning tasks.

| PROMPTS | Without Reasoning Guidance | | | | Reasoning Guidance | | | |
|---|---|---|---|---|---|---|---|---|
| | Test Img | +Reference Img | +Answered Pair | +Attribute Explanation | Test Img | +Reference Img | +Answered Pair | +Attribute Explanation |
| Phi-3.5-Vision [37] | 4.22 | 8.95 | 8.05 | 6.26 | 11.32 | 14.40 | 16.16 | 15.79 |
| LLaVA-OneVision [38] | 9.16 | 16.06 | 15.65 | 14.84 | 9.10 | 16.81 | 16.21 | 20.53 |
| DeepSeek-VL [39] | 8.16 | 22.68 | 20.03 | 12.65 | 14.02 | 20.31 | 24.62 | 25.45 |
| InternVL2.5 [40] | 15.79 | 18.82 | 23.30 | 17.16 | 15.63 | 22.35 | 24.47 | 23.73 |
| InternVL3 [40] | 15.46 | 17.90 | 22.44 | 17.91 | 15.81 | 21.98 | 23.58 | 17.05 |
| Qwen2.5-Omni [41] | 23.43 | 28.89 | 28.74 | 26.50 | 26.12 | 30.93 | 29.40 | 35.71 |
| Qwen2.5-VL [42] | 24.54 | 30.76 | 30.47 | 25.86 | 24.54 | 32.78 | 33.64 | 38.88 |
| Gemini2.5 Pro [43] | 24.23 | 30.09 | 29.39 | 25.55 | 25.68 | 30.61 | 32.30 | 35.88 |
| Claude4 Vision [44] | 23.47 | 29.67 | 30.04 | 25.58 | 24.94 | 33.00 | 34.13 | 36.32 |
| GPT-4o [45] | 26.06 | 31.56 | 32.28 | 27.52 | 27.79 | 34.17 | 39.71 | 38.56 |
| **Ours** | **80.86** | **82.99** | **83.24** | **82.67** | **81.35** | **83.11** | **82.87** | **84.03** |

a batch size of 64. We adopted the AdamW optimizer with a cosine annealing scheduler. The hyperparameters were configured as follows:(1) Learning rate: 1e-6 for RL (GRPO) training and 2e-5 for baseline SFT experiments;(2) Maximum input image size: 2,483,776 pixels;(3) Total GRPO training steps: 1500.

**Datasets and Metrics.** We evaluated both leading vision–language models (VLMs) and our proposed method on TriView2CAD, following the evaluation strategy outlined in Sec. 3.2. The synthetic test set was used to assess in-domain accuracy. Due to the scarcity of real-world data, all 3,000 real-world samples were reserved exclusively for testing—enabling evaluation of out-of-distribution (OOD) generalization in authentic CAD scenarios.

We design four distinct prompt configurations to probe their orthographic projection reasoning capabilities: 1) Test image-Only: Models receive only the target rasterized drawings; 2) Test image + Reference Image: In addition to the target drawings, models are given a reference image—rendered on the same geometry layer—with all dimension names to supply semantic alignment; 3) Test image + Answered Pair: Inputs consist of the target drawings plus a correctly paired raster image and its parameter vector, providing an exemplar mapping between visual features and quantitative parameters; 4) Test image + Attribute Explanation: Models are supplied with the target drawings alongside a detailed textual interpretation authored by professional engineers.

Given that orthographic projection reasoning involves complex multi-step inference and current pretrained VLMs have limited exposure to CAD drawings, we incorporated explicit reasoning guidance into each prompt format to encourage coherent reasoning trajectories. For each of the four prompt configurations, we outlined step-by-step instructions for task completion—structured to guide the model through the reasoning process systematically. The complete set of instructions for each format is provided in the supplementary materials.

## 5.1 Performance on in-domain test set

The experimental results (Tab. 3) demonstrate that our method significantly outperforms existing vision–language models (VLMs) on orthographic projection reasoning tasks across all prompt formats. Specifically, in the "Without Reasoning Guidance" setting, the "+Answered Pair" prompt achieves the highest accuracy of 83.24%. In the "With Reasoning Guidance" setting, our model reaches 84.03% accuracy with the "+Attribute Explanation" prompt. Notably, our method also delivers strong performance under the "Test Image Only" condition, achieving 80.86% accuracy—highlighting the effectiveness of our proposed approach.

It is worth noting that the performance improvement from "without reasoning guidance" to "with reasoning guidance" is not as significant. This is primarily because, through our CReFT approach, the model learns effective reasoning strategies via the difficulty-aware reward mechanism, thereby reducing the need for additional guidance to achieve further significant gains. Based on our benchmarking of the seven leading VLMs, we present several important findings, as follows: **1) This reasoning task remains a tough challenge for pretrained VLMs.** Overall, orthographic projection reasoning entails

not only reading textual dimensions and visual features, but also matching annotations to geometry primitives, counting structural elements, and computing composite parameters. **2) Different prompt formats influence the performance.** In both the Without Reasoning Guidance and With Reasoning Guidance settings, different prompt formats lead to varying degrees of performance improvement. **3)Introducing reasoning guidance yields consistent gains across all seven VLMs.** Introducing appropriate reasoning guidance significantly enhances the model's ability to perform reasoning tasks effectively. Further analysis regarding the three findings are presented in the supplementary materials.

## 5.2 Performance on real-world test set

We evaluate out-of-distribution (OOD) generalization using all 3,000 real-world samples, which reflect authentic CAD scenarios. The baseline model (Qwen2.5-VL) achieves 13.47% accuracy—lower than its 24.54% performance on synthetic data. This performance degradation is primarily attributed to the more complex structures and occlusion relationships in real-world data (Sec. 3.1), which elevates task difficulty. We also compare with a model trained via standard supervised fine-tuning (SFT), which achieves 36.15% accuracy. This improvement stems from substantial gains in Primitive Counting tasks. However, the model still struggles with Composite Parameter Computation—underscoring the limitations of conventional SFT. Our CReFT-based method achieves 46.67% accuracy, demonstrating its ability to effectively mitigate OOD challenges and deliver substantial performance gains on complex real-world data.

## 5.3 Ablation study

**With CoT VS without CoT.** The results from Tab. 2 reveal a significant improvement in performance when incorporating chain-of-thought (CoT) reasoning. Specifically, the accuracy for individual Task 3 increases from 46.16% to 74.15% after adding CoT , demonstrating a clear performance boost. When combining multiple training tasks, such as Task 1 + Task 3 and Task 2 + Task 3, the improvements are even more pronounced, with accuracies reaching 77.90%

Table 2: Ablation study results.

| METHOD | w/o CoT | with CoT |
|---|---|---|
| Task3 | 46.16 | 74.15 |
| Task1 + Task3 | 46.05 | 77.90 |
| Task2 + Task3 | 30.71 | 67.75 |
| Task1 + Task2 + Task3 | 46.24 | 81.35 |

and 67.75%, respectively. The most notable enhancement is observed in the combination of Task 1 + Task 2 + Task 3, where the accuracy increases from 46.24% to 81.35%. The primary source of these improvements lies in how CoT helps break down complex Composite Parameter Computation tasks into more manageable, incremental steps. The CoT mechanism, however, guides the model through each individual component of the computation, allowing for a step-by-step approach that improves the model's reasoning capability.

**Three training tasks.** Notably, in the CoT setting, combining Task 1 and Task 3 provides a performance boost, increasing accuracy from 74.15% to 77.90%. However, when Task 2 is added, the model's performance decreases. This decline can be attributed to the design of Task 2, which includes a substantial number of masked elements within the instruction data. By exposing the model to higher uncertainty, Task 2 forces the model to contend with incomplete or ambiguous information, which ultimately hinders its ability to perform effectively on the task. On the other hand, when all three tasks (Task 1 + Task 2 + Task 3) are used together, the model's reasoning ability is fully activated, resulting in a significant performance improvement. In contrast, in the without CoT setting, the model's performance is constrained by its inability to effectively handle multi-step reasoning tasks, where even the best-performing task combination achieves a maximum accuracy of only 46.24%.

## 6 Conclusion and Limitation

In this work, we address challenges in orthographic projection reasoning for Computer-Aided Design (CAD) by proposing CReFT-CAD, an innovative two-stage fine-tuning paradigm. We demonstrated that the integration of curriculum-driven reinforcement learning (RL) with difficulty-aware rewards effectively enhances the model's reasoning ability. To further advance the field, we introduce TriView2CAD, the first large-scale, open-source benchmark specifically designed for orthographic projection reasoning. With its comprehensive dataset of 200,000 synthetic and 3,000

real-world samples, each annotated with precise dimensions and accompanied by six interoperable data modalities. In conclusion, this work lays the groundwork for more robust, scalable solutions to CAD-related challenges, providing both a novel training paradigm and a rich dataset that can foster future research and innovation in the field.

**Limitation.** The accuracy on complex real-world scenarios has not yet reached a very high level. As such, the model's generalization ability remains a key area for further development. In future work, we plan to explore techniques to enhance the model's robustness and adaptability.

**ACKNOWLEDGMENT** This work is supported in part by NSFC Project (No. 62176061) and Joint Laboratory of Intelligent Construction Engineering Technology for Operating Railway Lines.

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

# A The complete reasoning guidance for each input format

**For the Test Image-Only format:**

- Identify all annotated numbers in the orthographic projection, including dimensions, spacing, and height.
- Interpret their meanings based on position, orientation, and surrounding context.
- Directly assign values to some parameters, count graphical elements for quantity parameters, and compute spacing parameters based on position.

**For the +Reference Image format:**

- Use the reference template to understand the correspondence between primitive and parameters, and identify annotated numbers in the target image.
- Assign parameters based on position and template primitive, counting graphical primitive for quantities.
- Calculate parameters based on positional relationships or template definitions, and output the full set of parameter values.

**For the +Answered Image format:**

- Learn the correspondence between structures and parameters from the example image.
- Identify and interpret annotated numbers in the target image, considering their position and similarity to the example.
- Assign values to parameters, count graphical primitive for quantities, calculate parameters, and apply necessary constraints to produce final results.

**For the +Attribute Explanation format:**

- Clarify parameter definitions and their geometric meanings, including component types and their properties.
- Identify annotated numbers in the orthographic projection and infer their corresponding parameters.
- Assign values to parameters, compute quantities, derive spacing values from positional relationships, and output the complete parameter set with necessary constraints.

# B Further analysis regarding the three findings

Table 3: Performance comparison of various VLMs on orthographic projection reasoning tasks.

| **PROMPTS** | **Without Reasoning Guidance** | | | | **Reasoning Guidance** | | | |
| --- | --- | --- | --- | --- | --- | --- | --- | --- |
| | Test Img | +Reference Img | +Answered Pair | +Attribute Explanation | Test Img | +Reference Img | +Answered Pair | +Attribute Explanation |
| Phi-3.5-Vision | 4.22 | 8.95 | 8.05 | 6.26 | 11.32 | 14.40 | 16.16 | 15.79 |
| LLaVA-OneVision | 9.16 | 16.06 | 15.65 | 14.84 | 9.10 | 16.81 | 16.21 | 20.53 |
| DeepSeek-VL | 8.16 | 22.68 | 20.03 | 12.65 | 14.02 | 20.31 | 24.62 | 25.45 |
| InternVL2.5 | 15.79 | 18.82 | 23.30 | 17.16 | 15.63 | 22.35 | 24.47 | 23.73 |
| InternVL3 | 15.46 | 17.90 | 22.44 | 17.91 | 15.81 | 21.98 | 23.58 | 17.05 |
| Qwen2.5-Omni | 23.43 | 28.89 | 28.74 | 26.50 | 26.12 | 30.93 | 29.40 | 35.71 |
| Qwen2.5-VL | 24.54 | 30.76 | 30.47 | 25.86 | 24.54 | 32.78 | 33.64 | 38.88 |
| Ours | **80.86** | **82.99** | **83.24** | **82.67** | **81.35** | **83.11** | **82.87** | **84.03** |

**1) Remains a Tough Challenge for pretrained VLMs.** Tab. 3 reports performance across four prompt formats without and with reasoning guidance. Overall, orthographic projection reasoning entails not only reading textual dimensions and visual features, but also matching annotations to geometry primitives, counting structural elements, and computing composite parameters. Consequently,

none of the off-the-shelf models fully masters this composite task. Qwen2.5-VL [46] achieves the highest accuracy—38.88% under reasoning guidance with the Test Image + Attribute Explanation prompt. This improvement stems primarily from its superior ability to parse geometry layers and read dimension labels. Crucially, its performance on reasoning-intensive parameters remains low. These findings underscore that orthographic projection CAD cannot be solved by prompting alone and requires dedicated fine-tuning strategy.

**2) Prompt Format–Dependent Performance Gains.** Appending an Attribute Explanation to the Test Image consistently boosts accuracy compared to the image-only baseline, with an average increase of 3 to 4 percentage points across models. This demonstrates that strong text encoders can leverage detailed, engineer-authored descriptions to guide complex geometric and numerical inferences. Similarly, providing a Reference Image or an Answered Pair results in comparable improvements, with a 4 to 5 percentage point increase in accuracy. These exemplars offer explicit visual-textual templates, simplifying the task of matching primitives to their corresponding semantic labels and dimensions. In contrast, using raw images forces models to address both perception and reasoning simultaneously, leading to the lowest performance. Collectively, these findings indicate that multimodal exemplars can enhance existing VLMs' ability to reason over orthographic projection. However, this further underscores that additional fine-tuning strategies are crucial to fully bridging the remaining performance gap.

**3)Introducing reasoning guidance yields consistent gains across all seven VLMs.** Introducing reasoning guidance demonstrates that step-wise reasoning guidance helps models better decompose the multi-step orthographic projection tasks. In the absence of reasoning guidance, visual exemplar prompts (+Reference Image and +Answered Pair) deliver the largest relative improvements. However, when reasoning guidance is added, the Attribute Explanation prompt shows the greatest uplift. It stems from reasoning guidance is textual form, which synergizes most effectively with textual inputs. Hence, models with stronger text encoders (e.g., DeepSeek-V1) exhibit disproportionately larger boosts. These results underscore the necessity of integrating structured reasoning guidance to advance orthographic projection CAD reasoning.

## C More performance on in-domain test set

Table 4: Performance comparison of training-free model, SFT model,and our GRPO-based model on orthographic projection reasoning tasks.

| Prompts | Without Reasoning Guidance | | | | Reasoning Guidance | | | |
|---|---|---|---|---|---|---|---|---|
| | Test Img | +Reference Img | +Answered Pair | +Attribute Explanation | Test Img | +Reference Img | +Answered Pair | +Attribute Explanation |
| Qwen2.5-VL | 24.54 | 30.76 | 30.47 | 25.86 | 24.54 | 32.78 | 33.64 | 38.88 |
| Qwen2.5-VL(SFT) | 76.33 | 79.50 | 77.12 | 78.64 | 76.42 | 76.78 | 77.20 | 80.30 |
| Ours | **80.86** | **82.99** | **83.24** | **82.67** | **81.35** | **83.11** | **82.87** | **84.03** |

Tab. 4 presents the performance in-domain test set. The first row (Qwen2.5-VL) represents a training-free baseline, which exhibits consistently poor performance across all settings, indicating the inherent difficulty of this reasoning task without any fine-tuning or adaptation. The second row (Qwen2.5-VL with SFT) demonstrates a significant performance improvement, particularly when provided with reference images, answered pairs, and attribute explanations. However, despite these gains, the model still suffers from notable drops in more complex reasoning tasks, especially those involving composite parameter computation. In contrast, our method consistently outperforms both baselines across all settings, achieving the highest accuracy in all prompt configurations, with particularly strong results under reasoning guidance. These results validate the effectiveness of our approach in handling complex geometric reasoning and highlight its robustness across both simple and compositional inference scenarios.

## D Failure cases and analysis

As illustrated in Fig. 5, we present four distinct failure cases, each highlighting different types of errors that can occur during the 3D model reconstruction process due to incorrect 2D parameterization.

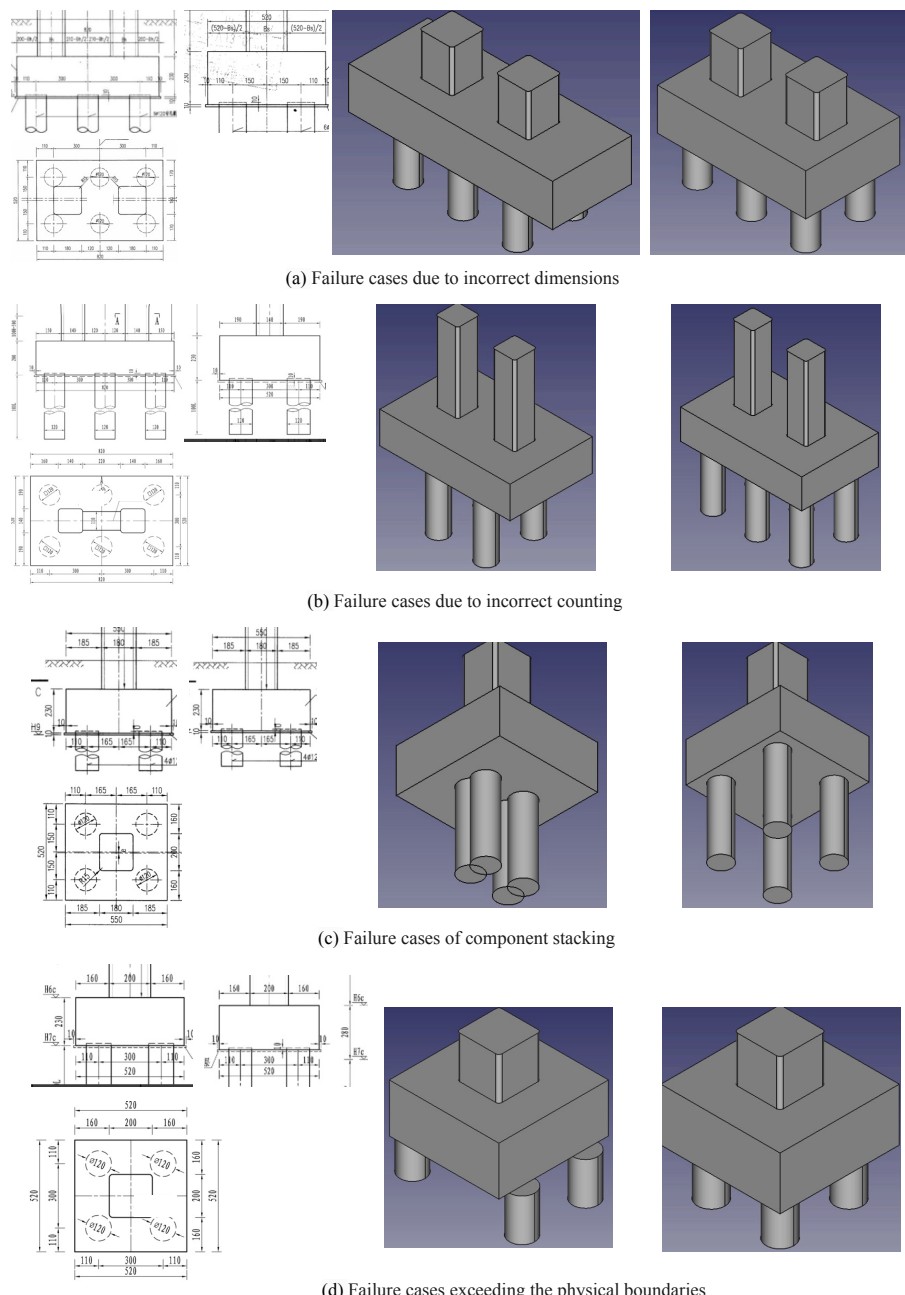

(a) Failure cases due to incorrect dimensions

(b) Failure cases due to incorrect counting

(c) Failure cases of component stacking

(d) Failure cases exceeding the physical boundaries

Figure 5: Four sets of failure cases. Each set consists of three parts: the left part shows the orthographic projection, the middle part presents the failed 3D model construction, and the right part illustrates the correctly constructed 3D model.

Fig. 5(a) demonstrates the error induced by incorrect dimensional parameters. Specifically, due to errors in the parameterization of certain dimensions, the constructed 3D model exhibits significant scaling issues, leading to a distorted and non-accurate representation. This failure emphasizes the critical importance of precise dimensional input when translating from 2D projections to 3D models, as even minor errors in size parameters can drastically affect the final output. Fig. 5(b) illustrates a failure caused by incorrect counting, which results in an incorrect number of primitives being identified and subsequently integrated into the 3D model. The discrepancy in the number of primitives

reflects the direct impact that improper counting and recognition of geometric elements can have on the success of the 3D reconstruction. In Fig. 5(c), we observe a stacking issue caused by errors in the calculation of composite parameters. The failure manifests as improper alignment of components in the final model, where parts of the 3D structure fail to align correctly with each other. Lastly, Fig. 5(d) shows a failure resulting from exceeding physical boundaries due to incorrect composite parameter calculations. In this case, the incorrect parameters lead to elements of the 3D model extending beyond the physical constraints or limits of the original design. Together, these four failure cases provide strong evidence that errors in the parameterization of 2D models directly lead to inaccuracies in 3D model reconstruction. Each case illustrates the cascading effect of parameter errors from the 2D representation to the final 3D model, further underscoring the necessity for robust and precise parameterization methods in 2D-to-3D model conversion processes.

