# OpenReview forum: "CReFT-CAD: Boosting Orthographic Projection Reasoning for CAD via Reinforcement Fine-Tuning"
_NeurIPS.cc/2025/Conference — NeurIPS 2025 poster_

### Official Review · Reviewer_LRf1 · 2025-06-04

**Clarity:** 3
**Significance:** 2
**Originality:** 2
**Rating:** 4
**Confidence:** 2

**Summary:**

The paper proposes a solution for using foundational models to help CAD in generating orthographic projection of the models, which is a crucial in the design pipeline, instead of directly going for 3D model generation. To this end, a new dataset is proposed and a fine-tuning strategy based on curriculum learning is presented. Experiments are done and compared with several foundational models. The results show the proposed new data with the fine-tuning strategy can achieve good results on the specific tasks.

**Questions:**

Following my comments below, I have a few questions below:

1. It seems that the primary contribution of the paper is the data. However, the majority of the data seems to be synthetic which is normally cleaner, easier to handle than real-world data. Since the root of the research is a real-world application. This might hurt the contribution of the paper. Also, from the provide visualization of the data, they seem to be restricted to a certain type of objects. Considering CAD is used almost in every industry, I wonder how diverse the data is. I am not an expert in CAD, so I will leave this to the domain expert to judge the value of the new data.

2. The proposed fine-tuning strategy is interesting. As an AI and machine learning conference, one natually asks the question how general the proposed method is when applied to other data and tasks. Currently, the curriculum design seems to be centered around the specific data and task. It would be good to discuss if there is fundamental difference between this specific curriculum design and other simiar research. I am asking for more experiments but only wonder if this new CAD task requires a categorically different curriculum design.

3. The OOD test on real-world data is highly appreciated. However the analysis is rather short. It is important for papers like this one to analyze what difficulties when facing real-world data. In what sense they are out of distribution? Is the worse performance mainly caused by the lower quality of real-world data, or the diversity? As a result, the paper did not seem to achieve the goal it claims to achieve in the beginning.

**Ethical Concerns:**

["NO or VERY MINOR ethics concerns only"]

**Final Justification:**

After the rebuttal, my questions are answered. Therefore, I am staying on the positive side.

**Limitations:**

yes

**Paper Formatting Concerns:**

no.

**Quality:**

2

**Strengths And Weaknesses:**

Strengths:

1. The paper is written clearly, easy to follow and understand.
2. The experiments can show certain promising aspects of the proposed methodology and the analysis on the comparative performances under different settings is helpful

Weaknesses:

1. There is an unclear difference between the proposed method and other similar methods in CAD, especially when it comes to the technical contribution.

2. The realism and diversityi of the data itself seem to limited.

3. Limited real-world applicability

Please see my comments below

---

> ### Author Rebuttal · Authors · 2025-07-30
>
> We sincerely thank Reviewer LRf1 for the thoughtful comments and for recognizing the clarity of presentation, the promising results, and the value of comparative analysis. Below we address your main concerns regarding (1) technical contribution beyond data, (2) dataset realism and diversity, and (3) generality and real-world applicability of our method.
>
>
> **W1 difference between the proposed method and other similar methods in CAD**
>
> SFT offers stable token-level supervision based on cross-entropy loss, optimizing the likelihood of the next token. While this is effective for mimicking labeled samples, it is insufficient for orthographic reasoning tasks, which require multi-step spatial inference and arithmetic computation—especially for counting parameters and composite calculation parameters.
>
> In contrast, we adopt a curriculum-based RL framework with task-level semantic rewards. Instead of optimizing per-token accuracy, we define goal-directed reward functions for full-task correctness:
> This method encourages the model to learn actual reasoning chains, not just memorize patterns. More importantly, RL allows effective learning without requiring large-scale high-quality annotations—an essential benefit in the CAD domain, where ground-truth labeling is both costly and domain-expert dependent.
>
> **W2&Q1 Diversity**
>
> Thank you for raising this important point. First, our choice of bridge pier drawings is motivated by their non-trivial geometric constraints, which inherently offer rich parameter spaces and meaningful semantic structures.
>
> Second, CAD drawings across domains exhibit substantial structural commonality. Tasks such as parameter extraction, component counting, and composite parameter calculations are ubiquitous subtasks in orthographic projection reasoning, regardless of the specific industry or object type. The three reinforcement learning tasks and their associated reward functions that we propose are directly transferable across different CAD domains sharing the orthographic paradigm. As such, although the current dataset may have limitations in geometric variability, it remains a solid foundation for developing robust and generalizable training strategies.
>
> Third, our dataset is a high-quality benchmark with detailed parameter annotations and real-world CAD drawings. It is extremely difficult to obtain, as it typically requires access to proprietary engineering documents and expert-level annotation. Its richness and annotation depth provide a solid foundation for developing and evaluating CAD reasoning task in realistic scenarios.
>
> Finally, since our approach does not rely on mimicking the original data distribution, but instead focuses on inducing true orthographic reasoning capability, we believe the model would maintain strong performance even if trained on other object types or visual styles. In fact, integrating multiple categories into the training process may introduce beneficial diversity, enhancing the model’s reasoning generality through exposure to broader spatial and semantic patterns.
>
> **W3&Q3 Limited real-world applicability**
>
> We fully acknowledge that achieving high accuracy in real-world orthographic projection reasoning, particularly under out-of-distribution (OOD) conditions, remains a significant and ongoing challenge.
>
> While the current performance on real-world data does not yet meet the stringent requirements of industrial deployment, it is nonetheless encouraging that our model consistently outperforms existing vision–language baselines under such demanding conditions. We believe this offers valuable insights and direction for future research within the community.
>
> As illustrated in Figure 3 of the main paper and Figure 1 of the supplementary material, real-world data in our benchmark presents three key challenges:
>
> Multi-part CAD Assemblies: A single real-world CAD drawing often contains all orthographic views of multiple components within one large drawing. This makes it non-trivial for the model to localize and reason about individual parameters. For visualization clarity, we cropped the most relevant regions for display in the paper, but the model itself must operate on the entire, cluttered canvas.
>
> Occlusions Across Components: Due to the multi-part layout, structural components often occlude each other (as shown in Fig. 3 top view). This creates visual interference that degrades spatial clarity and makes reasoning harder.
>
> Semantic Noise from Irrelevant Annotations: Because the drawing includes annotations for many unrelated parts, the model must disentangle relevant cues from semantically unrelated or misleading labels.
>
> These factors collectively make the real-world subset of TriView2CAD significantly harder than the synthetic benchmark. That said, we view our current gains on this data as a promising first step toward reliable CAD understanding, reflecting clear momentum rather than a shortcoming of the approach.
>
> As presented in Section D of the supplementary material, we provide visualizations and detailed analyses of failure cases encountered during testing on real-world data. These examples highlight specific challenges that remain open. In future work, we plan to further enhance our framework to address these issues more effectively and improve robustness under complex real-world conditions.
>
>
>
> **Q2  applied to other data and tasks**
>
> Based on our extensive experience with CAD-related tasks, we have found the following:
> The parameter space exhibits strong consistency across different types of CAD objects. Tasks such as parameterization, counting, and composite parameter computation are commonly found across a wide range of engineering domains.
>
> However, CAD drawings are used in many industries, and some sectors may indeed involve specialized domain requirements. In such vertical applications, customized parameter space design may further improve performance.
>
> The three reinforcement learning tasks and their associated reward functions that we propose are directly transferable across different CAD domains sharing the orthographic paradigm.
>
> Therefore, we believe our training paradigm is both robust and scalable, and can be effectively applied to new CAD fields—even those with greater geometric complexity or differing annotation standards.

---

> > ### Comment · Reviewer_LRf1 · 2025-08-03
> >
> > Thansk for the clarification.I will keep my score.

---

> > > ### Author Response · Authors · 2025-08-06
> > > **Appreciation for your positive evaluation**
> > >
> > > We sincerely appreciate your positive feedback and recognition of our work. Your insights have been invaluable in helping us refine our research, and we are grateful for the constructive guidance provided.
> > >
> > > we will work diligently to strengthen the technical rigor, clarity, and contribution of our work as outlined in your comments.
> > >
> > > Thank you again for your time and support.

---

### Official Review · Reviewer_jq8V · 2025-07-03

**Clarity:** 2
**Significance:** 2
**Originality:** 2
**Rating:** 2
**Confidence:** 5

**Summary:**

This paper performs orthographic projection reasoning in CAD workflows by introducing CReFT-CAD, a two-stage fine-tuning framework for vision–language models.The main idea is a Curriculum-driven Reinforcement Fine-Tuning (CReFT), a three-task curriculum (binary choice, multiple choice, chain-of-thought parameterization). They then perform supervised post-tuning of the output. The paper also introduces a TriView2CAD Benchmark, which is a new dataset of 200 K synthetic + 3 K real-world three-view samples.

**Questions:**

How sensitive are results to the chosen reward weights?
Have you analyzed which specific real-world drawing features (e.g., occlusions, annotation styles) cause most OOD failures? A breakdown by task type would be helpful.
What are the complex models in TriView2CAD dataset? Does it include more complex features than cylinders and blocks?

**Ethical Concerns:**

["NO or VERY MINOR ethics concerns only"]

**Final Justification:**

While I do acknowledge the authors effort in extending their evaluation to a more complicated dataset, the initial paper itself should have included this. In addition, this also requires a thorough evaluation of the approach on the expanded CAD dataset, which was missing from the initial submission. I still feel the work is in the right direction, but as is, it is a bit premature.

**Limitations:**

Authors acknowledge limited real-world accuracy and plan to improve generalization.
The dataset complexity seems very limited from the examples shown in the paper. I am willing to change this if the authors explain how diverse their dataset is.

**Paper Formatting Concerns:**

The quality of the figures could be improved

**Quality:**

2

**Strengths And Weaknesses:**

Strengths
+ The integration of RL followed by fine tuning seems to be reasonable.
+ TriView2CAD’ dataset can fill a critical gap in CAD reasoning datasets

Weaknesses
- Despite improvements, real-world accuracy remains under 50 %
- It is unclear what the different models make up the TriView2CAD dataset. Only a limited set of examples is shown in the paper making it unclear if the method will work on more complex CAD models with complex extruded features instead of just cylinders and blocks.

---

> ### Author Rebuttal · Authors · 2025-07-30
>
> We thank Reviewer jq8V for carefully reading our paper and for identifying important areas that warrant clarification and improvement. We appreciate your recognition of the value of combining reinforcement learning with post-tuning and the contribution of our TriView2CAD dataset to the CAD reasoning community.
> Below, we respond to each concern in detail and outline how we plan to address these points in the revised version.
>
> **W1&Q2 real-world accuracy**
>
> We fully acknowledge that achieving high accuracy in real-world orthographic projection reasoning, particularly under out-of-distribution (OOD) conditions, remains a significant and ongoing challenge.
>
> While the current performance on real-world data does not yet meet the stringent requirements of industrial deployment, it is nonetheless encouraging that our model consistently outperforms existing vision–language baselines under such demanding conditions. We believe this offers valuable insights and direction for future research within the community.
>
> As illustrated in Figure 3 of the main paper and Figure 1 of the supplementary material, real-world data in our benchmark presents three key challenges:
>
> Multi-part CAD Assemblies: A single real-world CAD drawing often contains all orthographic views of multiple components within one large drawing. This makes it non-trivial for the model to localize and reason about individual parameters. For visualization clarity, we cropped the most relevant regions for display in the paper, but the model itself must operate on the entire, cluttered canvas.
>
> Occlusions Across Components: Due to the multi-part layout, structural components often occlude each other (as shown in Fig. 3 top view). This creates visual interference that degrades spatial clarity and makes reasoning harder.
>
> Semantic Noise from Irrelevant Annotations: Because the drawing includes annotations for many unrelated parts, the model must disentangle relevant cues from semantically unrelated or misleading labels.
>
> These factors collectively make the real-world subset of TriView2CAD significantly harder than the synthetic benchmark. That said, we view our current gains on this data as a promising first step toward reliable CAD understanding, reflecting clear momentum rather than a shortcoming of the approach.
>
>
> **W2&Q3 Diversity**
>
> Thank you for raising this important point. First, our choice of bridge pier drawings is motivated by their non-trivial geometric constraints, which inherently offer rich parameter spaces and meaningful semantic structures.
>
> Second, CAD drawings across domains exhibit substantial structural commonality. Tasks such as parameter extraction, component counting, and composite parameter calculations are ubiquitous subtasks in orthographic projection reasoning, regardless of the specific industry or object type. The three reinforcement learning tasks and their associated reward functions that we propose are directly transferable across different CAD domains sharing the orthographic paradigm. As such, although the current dataset may have limitations in geometric variability, it remains a solid foundation for developing robust and generalizable training strategies.
>
> Third, our dataset is a high-quality benchmark with detailed parameter annotations and real-world CAD drawings. It is extremely difficult to obtain, as it typically requires access to proprietary engineering documents and expert-level annotation. Its richness and annotation depth provide a solid foundation for developing and evaluating CAD reasoning task in realistic scenarios.
>
> Finally, since our approach does not rely on mimicking the original data distribution, but instead focuses on inducing true orthographic reasoning capability, we believe the model would maintain strong performance even if trained on other object types or visual styles. In fact, integrating multiple categories into the training process may introduce beneficial diversity, enhancing the model’s reasoning generality through exposure to broader spatial and semantic patterns.
>
>
> It is also worth noting that domain specificity is a common design choice in this field. For example, in the recent work: CAD2Program[1], the entire training and evaluation pipeline is conducted on a single CAD domain—kitchen cabinet drawings. Their goal, which aligns with ours, is to test large-model reasoning capabilities under domain-specific constraints.
> In contrast, our contribution lies in introducing a reinforcement-learning-based, multi-task curriculum that strengthens generalization and structured reasoning.
>
> [1]Wang et al., From 2d cad drawings to 3d parametric models: A vision-language approach. In Proceedings of the AAAI Conference on Artificial Intelligence, volume 39, 7961–7969.
>
>
> **Q1 How sensitive are results to the chosen reward weights?**
>
> Thank you for raising this insightful point. We are happy to clarify how reward functions are applied in our framework.
>
> First, as noted in Equations (1)–(3) of the main paper, each sample in our dataset corresponds to a specific task type, and hence at training time, only one reward function is active per sample. Thus, there is no simultaneous interaction or weighting between different reward functions, and no reward balancing across task types is needed.
> Regarding reward value sensitivity (particularly in Eq. 2 and Eq. 3), we did not perform exhaustive ablations on individual reward magnitudes for the following reasons:
>
> Relative reward differences—not absolute values—drive optimization. The design goal of our reward functions is to distinguish high-quality from suboptimal outputs. As long as there are meaningful gaps in reward assignment, the policy gradient update encourages exploration along paths of relatively greater return. This mostly influences convergence speed, rather than final policy quality, especially after large-scale sampling.
>
> During RL updates, we compute advantage functions using normalized returns, which further attenuates the effect of raw reward magnitudes. This helps ensure training stability and makes performance less sensitive to specific numeric values.
> That said, we fully agree that studying reward value configurations is valuable. We are preparing an ablation experiment that tests representative reward schemes, including variations in the partial-credit structure for Eq. 2 and the difficulty-tiered scaling in Eq. 3.
> We will include these results in the revised version, and we appreciate your suggestion to further ground the method in principled analysis.

---

> ### Author Response · Authors · 2025-08-05
> **We are writing to kindly check if you have any remaining questions, concerns, or further suggestions.**
>
> We are writing to kindly check if you have any remaining questions, concerns, or further suggestions. If there are any aspects that require additional clarification or discussion, we would be more than happy to engage in further communication to address them promptly.

---

> > ### Comment · Reviewer_jq8V · 2025-08-05
> >
> > Thanks for answering my questions. However, the diversity of the data set is not clear. Hence I retain my rating.

---

> > > ### Author Response · Authors · 2025-08-05
> > > **Further Explanation on the Issue of Dataset Diversity**
> > >
> > > Dear Reviewer  jq8V,
> > >
> > > Thank you very much for your attention to the issue of dataset diversity in our research, which is crucial for enhancing the rigor and generalizability of the study. We have addressed your concerns in the rebuttal, but since they remain unresolved, we would like to provide further detailed explanations from the following aspects:
> > >
> > > **1.The dataset generation method is general, making the dataset scalable.**
> > >
> > > In Section 3.1 of the main text, we elaborated on our dataset generation method, which is applicable to any CAD drawings. Thus, the dataset can be extended to any fields according to users' needs. Meanwhile, the semantic structure of bridge pier drawings is clear, and the functions and geometric features of each component (such as pier body, bearing platform, foundation, etc.) have clear corresponding relationships, providing an ideal carrier for the model to learn the "parameter-semantic-constraint" associations in orthographic projection reasoning.
> > >
> > > **2.The contribution of our dataset to the community lies not only in synthetic data but also in real-world CAD drawings with detailed and accurate parameter annotations.**
> > >
> > > An important constraint that hinders the development of CAD-related work is the lack of high-quality datasets. Previous datasets do not include precise dimension annotations nor real-world design samples. Such datasets are extremely difficult to obtain, as they typically require access to proprietary engineering documents and expert-level annotation. The richness and annotation depth of our dataset provide a solid foundation for developing and evaluating CAD reasoning tasks in realistic scenarios.
> > >
> > > **3.CAD drawings across different domains share significant structural commonalities.**
> > >
> > > Regardless of the specific industry or object type, tasks such as parameter extraction, component counting, and composite parameter calculation are ubiquitous subtasks in orthographic projection reasoning. The three reinforcement learning tasks and their associated reward functions proposed by us can be directly transferred to different CAD domains that adopt the orthographic paradigm. Therefore, although the current dataset may have limitations in geometric diversity, it still lays a solid foundation for developing robust and generalizable training strategies.
> > >
> > > **4.Our method does not rely on mimicking the original data distribution but focuses on inducing true orthographic projection reasoning capabilities.**
> > >
> > > We believe that the model will maintain strong performance even when trained on other object types or visual styles. In fact, integrating multiple categories into the training process may introduce beneficial diversity, enhancing the model's reasoning generality by exposing it to a broader range of spatial and semantic patterns.
> > >
> > > **5.Domain specificity is a common scientific research choice in the CAD field.**
> > >
> > > Due to the wide range of industrial design fields covered by CAD, it is difficult to address too many types of problems in a single dataset. Therefore, conducting scientific research on a specific domain and then extending it to more domains is a common scientific research approach. For example, in the recent work CAD2Program [1], the entire training and evaluation pipeline is conducted on a single CAD domain—kitchen cabinet drawings. Their goal, which aligns with ours, is to test large-model reasoning capabilities under domain-specific constraints.
> > >
> > > [1] Wang et al., From 2d cad drawings to 3d parametric models: A vision-language approach. In Proceedings of the AAAI Conference on Artificial Intelligence, volume 39, 7961–7969.

---

> ### Comment · Area_Chair_3iYP · 2025-08-06
> **please provide a detailed answer**
>
> Dear jq8V (the authors are included in this one),
>
> Please read the authors' answer on the dataset diversity. Given that the majority of reviewers lean towards acceptance, you would need to provide a more detailed justification of your rating / assessment. Please also provide an answer to the rest of the authors' rebuttal points (real-world accuracy, sensitivity of the method to the reward weights). Are their answers satisfying these concerns or not?
>
> Thank you
> AC

---

> > ### Comment · Reviewer_jq8V · 2025-08-07
> >
> > My main concern is that the dataset only contains bridge pier drawings, while the paper is positioned as a general CAD method. In traditional component CAD, such primitives form a very small percentage (<10%). Since the approach has only been tested on this narrow subset, the utility of this work as generalizable to the whole of CAD is questionable. If the paper is on RL for Bridge Pier Drawings, the focus of the paper gets so narrow that it would not be suitable for a general audience, such as NeurIPS. While the authors claim that the method is generalizable, without actual proof that it works on more general CAD drawings, it is difficult to recommend the paper.

---

> > > ### Author Response · Authors · 2025-08-08
> > > **Response to Reviewer jq8V**
> > >
> > > Dear Reviewer jq8V,
> > >
> > > Thank you very much for carefully reading our rebuttal and comments, as well as for your time and valuable suggestions. We believe that further elaborating on the generalization of our data and method will be highly beneficial to the paper.
> > >
> > > Regarding the issue of **dataset diversity**, we would like to kindly reiterate that, as mentioned in points 1 and 3 of the comments, the data construction method we proposed is applicable to any type of CAD model. Moreover, CAD drawings across different domains share significant structural commonalities.
> > >
> > > Concerning **the generalization of our method in CAD tasks**, we supplement an experiment to demonstrate this. Firstly, we obtained data from the open-source dataset DeepCAD [1]. The DeepCAD dataset is a computer-aided design (CAD) models dataset, consisting of 197k distinct 3D models. Since it lacks accurately annotated orthographic projection data, we implement the following data processing :
> > >
> > > 1.  Use FreeCAD to process Step models and obtain DXF files of orthographic views.
> > >
> > > 2.  Use AutoCAD to annotate some dimensions in the DXF files.
> > >
> > > 3.  Render the DXF files into raster images.
> > >
> > > Due to the fact that performing the above steps and conducting training and testing with the same amount of data as in the main text is extremely time-consuming and cannot be completed during the discussion period, we annotate 2,500 samples, with 2,000 used for training and 500 for testing. To prevent model underfitting, we increased the training epochs to 3. As in the settings in the main text, the parameter space design includes recognition parameters, counting parameters (such as counting circles or rectangles in the drawings), and composite calculation parameters (such as calculating the perimeter of the bottom edge of the part) that can be widely used for all CAD models. The experiment's prompt was set to "without Reasoning Guidance + test img".
> > >
> > > | Method            | Accuracy (%) |
> > > | ----------------- | ------------ |
> > > | InternVL3 \[2]    | 28.6         |
> > > | Qwen2.5-Omni \[3] | 33.2         |
> > > | Qwen2.5-VL \[3]   | 34.0         |
> > > | SFT               | 51.8         |
> > > | Ours              | 65.6         |
> > >
> > > The upper part of the table directly tests the performance of existing excellent VLMs without fine-tuning. We can draw the same conclusion as in the main text: the orthographic projection reasoning task is challenging for pre-trained models and requires targeted fine-tuning design. The lower part of the table reports that using our annotated training data, Supervised fine-tuning (SFT) can achieve an accuracy of 51.8%, which is a significant improvement compared to the pre-trained models. After applying our Curriculum-driven Reinforcement Fine-tuning method, the performance is further improved. This is mainly because, for more complex parameter spaces, our method does not rely on mimicking the original data distribution but focuses on inducing true orthographic projection reasoning capabilities. Thus, it can be proven that on the DeepCAD dataset where all models are different, our method can achieve impressive performance improvements, demonstrating the generalization of our method.
> > >
> > > In the above rebuttal, we have responded to all your concerns. We kindly ask whether the other responses have resolved your concerns. We hope that this comment can address the only remaining concern you have.
> > >
> > > Thank you again for your reply.
> > >
> > > References:
> > >
> > > [1] Wu R, Xiao C, Zheng C. Deepcad: A deep generative network for computer-aided design models[C]//Proceedings of the IEEE/CVF International Conference on Computer Vision. 2021: 6772-6782.
> > >
> > > [2] Daya Guo, Dejian Yang, Haowei Zhang, Junxiao Song, Ruoyu Zhang, et al. Qwen2.5-omni technical report. arXiv preprint arXiv:2503.20215, 2025.
> > >
> > > [3] Qwen Team. Qwen2.5-vl, January 2025. URL https://qwenlm.github.io/blog/qwen2.5-vl/.

---

### Official Review · Reviewer_M7s3 · 2025-07-03

**Clarity:** 2
**Significance:** 3
**Originality:** 3
**Rating:** 4
**Confidence:** 4

**Summary:**

The paper introduces CReFT-CAD, a two-stage fine-tuning paradigm for orthographic projection reasoning in the context of Computer-Aided Design (CAD). The method integrates curriculum-driven reinforcement learning with difficulty-aware rewards, followed by a phase of supervised post-tuning to bolster instruction following and semantic extraction. To support evaluation, the authors present TriView2CAD, a large-scale benchmark containing 200,000 synthetic and 3,000 real-world samples, each linked to six interoperable data modalities with precise dimension annotations. Experiments benchmark leading vision-language models (VLMs) and demonstrate that CReFT-CAD yields improved accuracy and out-of-distribution generalization on orthographic projection reasoning tasks.

**Questions:**

1: Could authors provide further analysis of the curated datasets (e.g., the diversity of shapes/objects, etc)?

2: How would the performance of industry-level VLMs (such as GPT-4o, Gemini 2.5 flash) be on the proposed benchmark?

**Ethical Concerns:**

["NO or VERY MINOR ethics concerns only"]

**Final Justification:**

I recommend accepting paper because of its contribution to data collection. I hold the opinion that the biggest contribution of this paper is a large-scale CAD dataset to assist the community in verifying ideas on CAD model understanding. And, the authors' replies address most of my concerns about evaluation. Therefore, I recommend acceptance.

**Limitations:**

Yes.

**Paper Formatting Concerns:**

N/A.

**Quality:**

3

**Strengths And Weaknesses:**

Strengths:

1: The introduction of TriView2CAD is a meaningful addition to the field. The benchmark is described in detail and includes synthetic and real-world samples across six data modalities, supporting a wide spectrum of CAD tasks. This addresses long-standing challenges in the lack of open, precisely annotated datasets for orthographic projection reasoning.

2: The proposed CReFT-CAD leverages a curriculum-driven reinforcement learning approach with customized, difficulty-aware reward functions. This approach is designed to systematically build up reasoning ability, followed by supervised post-tuning targeted at enhancing instruction. The methodology is well described and is tightly aligned to the practical needs of the CAD workflow (design, manufacturing, simulation).

3: Extensive experiments are conducted, benchmarking seven VLM baselines plus the proposed method. The results show CReFT-CAD's clear superiority over existing VLM baselines, especially on complex tasks and out-of-distribution (OOD) real-world scenarios.

Weaknesses:

1: **Analysis of Prompt Engineering.** Several prompt configurations are presented, both with and without reasoning guidance. While it is shown that reasoning guidance helps, it is not always clear if the observed gains are attributable to the underlying model improvements or just to the engineered prompt. In addition, the evaluation assumes a fixed set of prompt templates, which could constrain generalizability.

2: **Insufficient Analysis of Dataset Diversity.** The paper does not provide a thorough analysis of the characteristics of the proposed dataset, TriView2CAD. Based on the visualizations provided, many CAD models seem structurally similar, often consisting of repetitive patterns (e.g., cylinder-cuboid-cylinder arrangements). Such structural uniformity might limit the diversity of the benchmark, potentially inflating performance metrics or restricting the robustness evaluation of the models. Therefore, it is crucial to systematically assess and explicitly discuss the dataset's geometric and structural diversity.

3: **Evaluation of Industrial-Level VLMs.** Although the paper benchmarks various existing vision-language models (VLMs), it neglects to include current industrial-grade or commercially deployed VLMs in its evaluations. Given that the practical motivation is closely tied to industrial CAD applications, testing state-of-the-art industrial VLM would be highly valuable. Such analysis could provide clearer insights into how the proposed method compares against real-world industry standards and further demonstrate the practical significance of the proposed framework.

---

> ### Author Rebuttal · Authors · 2025-07-30
>
> We sincerely thank Reviewer M7s3 for the constructive feedback and for acknowledging the significance of both our CReFT-CAD framework and the TriView2CAD benchmark. We appreciate your recognition of the real-world motivation, the alignment to CAD workflow stages, and the model’s superior performance on complex and out-of-distribution (OOD) tasks.
> Below we address your thoughtful critiques regarding prompt engineering analysis, dataset diversity, and comparisons to industrial-grade models.
>
> **W1 Analysis of Prompt Engineering**
>
> As detailed in Lines 300–305 of the main paper and further elaborated in Appendix B.3, we conducted a systematic study of prompt engineering from two perspectives:
> (1) the presence or absence of reasoning guidance, and
> (2) the effect of different input formatting styles, including attribute explanations and structural text augmentation.
> While we observe that reasoning-guided prompts lead to performance improvements across all models, CReFT-CAD consistently outperforms all baseline VLMs under identical prompt templates. This strongly suggests that the gains stem primarily from the enhanced reasoning ability acquired during RL fine-tuning, not merely from the prompt structure.
>
> That said, we fully acknowledge the critical role of prompt design in VLM adaptation. Our experiments are meant to shed light on this often-understudied factor by providing comparative insights into how prompting strategies affect both our method and existing VLM baselines. We hope these results offer meaningful guidance to the community.
>
> Importantly, the prompt templates are not hard-coded or universal—in practice, reasoning guidance provided by domain experts varies substantially across CAD categories. Moreover, we found that adding attribute-level textual explanations, such as descriptions of component functions or constraints, significantly enhances performance on orthographic-view reasoning tasks. The chosen input modality (text, image, etc.) should be seen as a flexible reference point rather than a rigid constraint.
>
> **W2 &Q1 Insufficient Analysis of Dataset Diversity**
>
> Our selection of prefabricated bridge pier drawings is grounded in their geometric complexity. Despite belonging to a specific subdomain, these structures exhibit rich parametric relationships and semantic constraints that pose meaningful challenges for orthographic projection reasoning.
>
> Importantly, many CAD tasks across diverse domains frequently involve similar cognitive subtasks, such as parameter extraction, component enumeration, and multi-view composite parameter inference.
>
> The curriculum tasks we designed for reinforcement learning are not tailored to a specific object type, but rather capture generalizable reasoning patterns that are widely applicable to other CAD categories. Since these tasks are grounded in fundamental principles of orthographic projection reasoning, they can be readily transferred and reused across domains that follow similar design logic.
>
> We also emphasize that, our dataset is a high-quality benchmark with detailed parameter annotations and real-world CAD drawings. It is extremely difficult to obtain, as it typically requires access to proprietary engineering documents and expert-level annotation. Its richness and annotation depth provide a solid foundation for developing and evaluating CAD reasoning task in realistic scenarios.
>
> Lastly, the core strength of our method lies in learning the reasoning process itself, rather than fitting to any fixed data distribution. This property makes the approach generalizable to other CAD categories. We further posit that expanding the dataset to include multiple CAD domains will likely further enhance model robustness by encouraging the learning of abstract and transferable inference patterns.
>
> **W3&Q2Evaluation of Industrial-Level VLMs.**
>
> This is an excellent and important suggestion. We have extended our evaluation to include several industry-level VLMs, specifically Gemini 2.5 Pro, Claude 4 Vision, and GPT-4o. The results of these evaluations have been compiled and presented in the same format as Table 1 in the main paper, for consistency and comparison.
> We will integrate these results into the revised version of the paper, along with additional analysis of each model’s strengths and limitations on orthographic projection reasoning tasks.
>
> | Gemini 2.5 Pro   | 24.23 | 30.09 | 29.39 | 25.55 | 25.68 | 30.61 | 32.30 | 35.88 |
>
> | Claude 4 Vision  | 23.47 | 29.67 | 30.04 | 25.58 | 24.94 | 33.00 | 34.13 | 36.32 |
>
> | GPT-4o &nbsp;&nbsp;&nbsp;   &nbsp;&nbsp;&nbsp;      &nbsp;&nbsp;&nbsp;  &nbsp;   | 26.06 | 31.56 | 32.28 | 27.52 | 27.79 | 34.17 | 39.71 | 38.56 |

---

> ### Author Response · Authors · 2025-08-05
> **We are writing to kindly check if you have any remaining questions, concerns, or further suggestions.**
>
> We are writing to kindly check if you have any remaining questions, concerns, or further suggestions. If there are any aspects that require additional clarification or discussion, we would be more than happy to engage in further communication to address them promptly.

---

> ### Comment · Reviewer_M7s3 · 2025-08-06
>
> Thanks for the authors' reply. Most of my concerns have been addressed. I decide to keep my rating.

---

> > ### Author Response · Authors · 2025-08-06
> > **Appreciation for your positive evaluation**
> >
> > We sincerely appreciate your positive feedback and recognition of our work. Your insights have been invaluable in helping us refine our research, and we are grateful for the constructive guidance provided.
> >
> > we will work diligently to strengthen the technical rigor, clarity, and contribution of our work as outlined in your comments.
> >
> > Thank you again for your time and support.

---

### Official Review · Reviewer_Kh3L · 2025-07-03

**Clarity:** 3
**Significance:** 3
**Originality:** 3
**Rating:** 4
**Confidence:** 3

**Summary:**

This paper aims to address a long-standing challenge in the field of CAD: enabling AI models to perform accurate reasoning on orthogonal projection drawings. The authors point out that existing methods, whether traditional 3D reconstruction or simple SFT, suffer from issues such as insufficient accuracy, lack of editability, and poor generalization ability.

To tackle these problems, the paper proposes two major core contributions:

CReFT-CAD Method: An innovative two-stage fine-tuning paradigm. The first stage employs curriculum-driven RL, cultivating the model's deep reasoning capabilities through tasks ranging from easy to difficult and a difficulty-aware reward mechanism. The second stage enhances the model's instruction following and interactive capabilities through supervised post-fine-tuning.

TriView2CAD Dataset: The first large-scale, open-source benchmark specifically designed for this task. It comprises 200,000 synthetic samples and 3,000 real-world samples, featuring precise dimension annotations and six interoperable data modalities, providing valuable resources and evaluation standards for research.

Experimental results demonstrate that CReFT-CAD significantly outperforms several leading VLMs on the TriView2CAD benchmark and exhibits stronger generalization ability on real-world OOD samples.

**Questions:**

* What major challenges do you foresee in applying the CReFT-CAD framework to other CAD domains? Would differences in parameter space and increased geometric complexity necessitate significant modifications to the curriculum or reward functions?

* Could you provide specific examples of incorrect answers given by the model on real-world datasets? Analyzing these failure cases could help us better understand the limitations of the current method and directions for future improvements.

**Ethical Concerns:**

["NO or VERY MINOR ethics concerns only"]

**Final Justification:**

The authors have effectively transferred a mature pipeline from the LLM domain—combining Supervised Fine-Tuning with curriculum-based Reinforcement Learning to the field of CAD. This application demonstrates a degree of novelty, but I find the innovation to be somewhat limited, as it is primarily an application of an established methodology to a new domain rather than a fundamental advance.
For this reason, I will maintain my original rating. The paper is technically solid, and the reasons to accept outweigh the reasons to reject, but the limited methodological novelty makes it a borderline case.

**Limitations:**

yes

**Quality:**

3

**Strengths And Weaknesses:**

**Strenghths:**
* The "orthogonal projection reasoning" problem addressed by this paper is one of immense value in the industry, yet remains insufficiently resolved within the deep learning community. The authors provide a compelling motivation for their proposed approach.

* The CReFT-CAD method's design, which mimics the human cognitive learning process by progressing from easier to more challenging tasks, effectively contributes to the stable establishment of the model's reasoning capabilities.

* On benchmark tests, CReFT-CAD's accuracy significantly surpasses all baseline models, unequivocally demonstrating the effectiveness of their methodology.

**Weaknesses**
* All experiments in the paper are concentrated on a specific CAD category: prefabricated bridge piers. While this serves as an excellent starting point, its parameter space and geometric complexity remain relatively limited.

* The paper claims that its method enhances "reasoning ability," which is demonstrated through its task design and performance metrics. However, the analysis regarding the precise nature of the reasoning capabilities the model has acquired is not yet sufficiently in-depth.

---

> ### Author Rebuttal · Authors · 2025-07-30
>
> We sincerely thank Reviewer Kh3L for the thorough and constructive review. We are grateful for your positive assessment of our motivation, technical design, and empirical contributions. We especially appreciate your recognition of the industrial value of orthographic projection reasoning, the cognitive structure of our curriculum-based RL method, and the strong performance of our model on the TriView2CAD benchmark.
> Below, we respond in detail to the two main concerns and two insightful questions you raised.
>
> **W1&Q1 CAD category limited**
>
> Our selection of prefabricated bridge pier drawings is grounded in their geometric complexity. Despite belonging to a specific subdomain, these structures exhibit rich parametric relationships and semantic constraints that pose meaningful challenges for orthographic projection reasoning.
>
> Importantly, many CAD tasks across diverse domains frequently involve similar cognitive subtasks, such as parameter extraction, component enumeration, and multi-view composite parameter inference.
> The curriculum tasks we have designed for reinforcement learning are not tailored to a specific object type, but rather capture generalizable reasoning patterns that are widely applicable to other CAD categories. Since these tasks are grounded in fundamental principles of orthographic projection reasoning, they can be readily transferred and reused across domains that follow similar design logic.
>
> We also emphasize that, our dataset is a high-quality benchmark with detailed parameter annotations and real-world CAD drawings. It is extremely difficult to obtain, as it typically requires access to proprietary engineering documents and expert-level annotation. Its richness and annotation depth provide a solid foundation for developing and evaluating CAD reasoning task in realistic scenarios.
>
> Lastly, the core strength of our method lies in learning the reasoning process itself, rather than fitting to any fixed data distribution. This property makes the approach generalizable to other CAD categories. We further posit that expanding the dataset to include multiple CAD domains will likely further enhance model robustness by encouraging the learning of abstract and transferable inference patterns.
>
> **W2 reasoning ability**
>
> From a quantitative analysis perspective, we have initially conducted experiments using the same training data but applied only a supervised fine-tuning (SFT) strategy. Under evaluation settings analogous to those in Table 1 of the main paper, this SFT-only baseline achieved an overall accuracy of 80.3%, with 85.4% on recognition parameters, 97.1% on counting parameters, and 66.8%on composite calculation parameters. In contrast, our reinforcement learning–based training method achieved 84.03% overall accuracy, with 87.1% on recognition parameters, 97.3% on counting parameters, and a significantly improved 74.3% on composite calculation parameters. This comparison clearly demonstrates that reinforcement learning brings substantial gains, particularly for tasks requiring multi-step inference and reasoning, such as composite parameter calculation. These results provide strong empirical evidence that our curriculum-based RL strategy effectively endows the model with enhanced reasoning capabilities beyond what is achievable through pattern imitation via SFT alone.
>
> From a qualitative analysis perspective, traditional supervised fine-tuning (SFT) offers token-level supervision via cross-entropy loss, optimizing the next-token likelihood. While this approach enables effective pattern replication from annotated samples, it typically lacks the capacity to instill genuine reasoning. In contrast, our framework applies curriculum-based reinforcement learning, with task-level, semantically meaningful rewards. Rather than optimizing for local token-level correctness, we design goal-driven reward functions that evaluate whether the model completes the entire task correctly. This structure encourages the learning of full reasoning chains rather than superficial pattern memorization. Through this setup, the model acquires multi-step inference abilities necessary for solving complex CAD tasks like parameter counting, component relationships, and dimensional composition.
>
>
> **Q2 Failure cases**
>
> As shown in Appendix section D, we provide several representative failure cases along with analysis.

---

> > ### Comment · Reviewer_Kh3L · 2025-08-06
> > **Official Comment by Reviewer Kh3L**
> >
> > I thank the authors for their rebuttal, which has successfully addressed my concerns. They have effectively transferred a mature pipeline from the LLM domain—combining Supervised Fine-Tuning with curriculum-based Reinforcement Learning to the field of CAD. This application demonstrates a degree of novelty, but I find the innovation to be somewhat limited, as it is primarily an application of an established methodology to a new domain rather than a fundamental advance.
> > For this reason, I will maintain my original rating. The paper is technically solid, and the reasons to accept outweigh the reasons to reject, but the limited methodological novelty makes it a borderline case.

---

> > > ### Author Response · Authors · 2025-08-06
> > > **Appreciation for your positive evaluation**
> > >
> > > We sincerely appreciate your positive feedback and recognition of our work. Your insights have been invaluable in helping us refine our research, and we are grateful for the constructive guidance provided.
> > >
> > > we will work diligently to strengthen the technical rigor, clarity, and contribution of our work as outlined in your comments.
> > >
> > > Thank you again for your time and support.

---

> ### Author Response · Authors · 2025-08-05
> **We are writing to kindly check if you have any remaining questions, concerns, or further suggestions.**
>
> We are writing to kindly check if you have any remaining questions, concerns, or further suggestions. If there are any aspects that require additional clarification or discussion, we would be more than happy to engage in further communication to address them promptly.

---

### Official Review · Reviewer_8yVX · 2025-07-06

**Clarity:** 2
**Significance:** 3
**Originality:** 4
**Rating:** 4
**Confidence:** 4

**Summary:**

This paper explores the problem of reasoning 3D CAD shapes with canonical orthographic projection images from multiple views. First, a large scale dataset is prepared with annotated drawings rendered in orthographic views and paired 3D shape data using parametric variations of constrained pier spacing configurations. Then, four tasks of varying difficulties are designed to probe if vision language models can truly reason by establishing correspondences between the orthographic views and the 3D shape. Using reinforcement learning and instruction tuning, the GOT-OCR2.0 model is aligned on 4 tasks of increasing complexity using curriculum learning and demonstrated to reason significantly better than off the shelf. Vision language models.

**Questions:**

- Can the authors motivate better on why reinforcement  learning was required and supervised finetuning with standard next token prediction loss wasn’t sufficient? This design decision should be explained and it would be valuable for readers.
- Can the method be tested on examples that are not piers? This is crucial to understand how well the training scheme helps the model to truly understand orthographic view images of CAD shapes.
- Can the authors touch upon real world CAD problems that would benefit from the improvements in reasoning achieved in this work? This would solidify the impact of this work and help readers understand the motivation of this paper.

**Ethical Concerns:**

["NO or VERY MINOR ethics concerns only"]

**Final Justification:**

I stand by the original score. The authors have addressed most of my concerns in the rebuttal and I expect them to revise the final paper accordingly. But even in my original review, I was tending towards accepting the paper, so I will stick with the same score.

**Limitations:**

Yes

**Quality:**

3

**Strengths And Weaknesses:**

Strengths:

- the paper tackles an important and under-explored problem of reasoning 3D CAD shapes from orthographic drawings. Drawings are indeed a key component in CAD design and manufacturing workflows, and not many previous works have explored techniques to improve reasoning of vision language models on drawings.
- the large scale dataset with paired data containing drawings, their dimensions, and STEP files are useful for the community to build upon, and is a good contribution.
- the proposed method clearly works well as demonstrated by strong quantitative results.

Weaknesses:

I don’t see any major weaknesses in the paper with regards to problem scoping or technical contributions. But a few points:
- The exposition could be improved. It is not clear why reinforcement learning is required for this problem. The data preparation for training and specifics of how the model output connects with the loss function must be made more clear.
- both the synthetic and real world data for testing are parametrically varied piers. This narrows the scope quite drastically and generalization outside of this distribution was not demonstrated.
- the practical impact of better reasoning orthographic view reasoning achieved in this work is not clearly demonstrated by showcasing improvements in specific downstream applications.

---

> ### Author Rebuttal · Authors · 2025-07-30
>
> We sincerely thank Reviewer 8yVX for the thoughtful evaluation and for recognizing the technical soundness, originality, and relevance of our work. We appreciate your positive remarks on the importance of orthographic reasoning in CAD workflows, the value of our large-scale dataset, and the strong quantitative results. Below, we address your comments in detail.
>
> **W1&Q1: Why reinforcement learning is required**
>
> Orthographic projection reasoning presents fundamentally different challenges compared to conventional CAD tasks such as primitive detection or parametric annotation. While the latter focus on identifying individual geometric entities and retrieving associated dimensions, orthographic reasoning requires a holistic semantic understanding of multi-view technical drawings. This includes tasks such as cross-view correspondence, and performing composite parameter calculations that span multiple projections and demand multi-step spatial and logical inference.
>
> Such complexity cannot be effectively addressed through supervised learning alone, which primarily drives models to imitate training distributions rather than truly reason. Specifically, Supervised Fine-Tuning (SFT) provides stable token-level optimization via cross-entropy loss, which is useful for formatting and mimicry, but inherently limited in fostering the higher-order reasoning skills required for orthographic tasks.
>
> To overcome this limitation, we introduce a curriculum-based Reinforcement Learning (RL) framework with task-level, semantically grounded reward functions. Rather than optimizing token-level accuracy, our method defines goal-directed rewards based on full-task correctness, covering binary decisions, multiple-choice selection, and chain-of-thought parameterization.
> This approach enables the model to move beyond surface-level pattern recognition by cultivating structured reasoning chains tailored to the logic of orthographic projection.  Moreover, by optimizing directly for task-level correctness through semantically meaningful rewards, the method reduces reliance on large-scale, high-quality annotations, which are often expensive to obtain and require significant domain expertise in the CAD field.
>
> By combining curriculum-driven RL with SFT post-tuning, our framework balances reasoning capability with instruction-following fluency, making it well-suited for complex, industrial-grade CAD understanding.
>
> **W1: Clarifying Data-Training-Loss Connections**
>
> As described in Section 3.1 of the main paper, we have detailed the construction of our dataset, including the synthetic data generation pipeline.
> As described in Lines 192–220 of the main paper, during the reinforcement learning phase, we have constructed 160,000 image–instruction pairs from the synthetic subset of the TriView2CAD dataset. Each prompt contains an orthographic projection view alongside its complete set of 15 parametric key–value pairs. Based on our three designed learning tasks, we apply corresponding reward functions to compute relative advantages for policy optimization. This structured curriculum enables the model to progressively activate and refine its orthographic reasoning capabilities, rather than relying on superficial feature extraction.
> In the Supervised Post-Tuning phase, we have further refined the model using 20,000 instruction–response pairs, aiming to strengthen its instruction-following abilities in a more controlled and aligned setting. Beyond generic response formatting, this stage is specifically designed to equip the model with capabilities tailored to real-world CAD tasks. Based on our deep understanding of practical industrial workflows, we have structured the supervision to support four essential task types: producing a complete parameter list from multi-view drawings, validating parameter correctness against reference values, matching corresponding elements across orthographic views, and counting components embedded within complex assemblies. These tasks closely reflect operational needs in engineering design, simulation, and quality assurance, thus ensuring the model’s outputs are not only syntactically correct but also functionally relevant and deployable in real CAD scenarios.
>
>
> **W2&Q2 Limited Task Domain**
>
> First, our choice of bridge pier drawings is motivated by their non-trivial geometric constraints, which inherently offer rich parameter spaces and meaningful semantic structures.
>
> Second, CAD drawings across domains exhibit substantial structural commonality. Tasks such as parameter extraction, component counting, and composite parameter calculations are ubiquitous subtasks in orthographic projection reasoning, regardless of the specific industry or object type. The three reinforcement learning tasks and their associated reward functions that we propose are directly transferable across different CAD domains sharing the orthographic paradigm. As such, although the current dataset may have limitations in geometric variability, it remains a solid foundation for developing robust and generalizable training strategies.
>
> Third, our dataset is a high-quality benchmark with detailed parameter annotations and real-world CAD drawings. It is extremely difficult to obtain, as it typically requires access to proprietary engineering documents and expert-level annotation. Its richness and annotation depth provide a solid foundation for developing and evaluating CAD reasoning task in realistic scenarios.
>
> Finally, since our approach does not rely on mimicking the original data distribution, but instead focuses on inducing true orthographic reasoning capability, we believe the model would maintain strong performance even if trained on other object types or visual styles. In fact, integrating multiple categories into the training process may introduce beneficial diversity, enhancing the model’s reasoning generality through exposure to broader spatial and semantic patterns.
>
>
> **W3&Q3:Downstream applications**
>
> As shown in Lines 527–537 of the main paper, we have conducted evaluations on real-world data under out-of-distribution (OOD) settings. Even in these challenging scenarios, our method demonstrate substantial performance gains, highlighting its robustness and generalization capability beyond the synthetic training distribution.
>
> As presented in Section D of the supplementary material, we have provided visualizations and detailed analyses of failure cases encountered during testing on real-world data. These examples highlight specific challenges that remain open. In future work, we plan to further enhance our framework to address these issues more effectively and improve robustness under complex real-world conditions.

---

> > ### Comment · Reviewer_8yVX · 2025-08-07
> >
> > Thanks for the clarifications in the rebuttal.
> >
> > For W1 and W1&Q1, adding some of these points to the main paper would be helpful.
> >
> > For W3&Q3, I appreciate that the authors had evaluated the method on real world & OOD data, but my concern was more the impact that this reasoning capability brings to designers using CAD software. The paper does not get into the practical benefits of having orthographic reasoning capabilities, so some commentary on that in the introduction would be useful to help motivate your method better.

---

> > > ### Author Response · Authors · 2025-08-08
> > > **Appreciation for your suggestions**
> > >
> > > Dear Reviewer 8yVX,
> > >
> > > We sincerely appreciate your recognition of our work and the revision suggestion. Your suggestions can effectively help us clarify the motivation of our work and our method design more clearly, which are of great benefit to improving our paper.
> > >
> > > We will follow your advice and integrate some points comparing SFT and RL into the main text. Additionally, we will add the practical benefits of our method for real CAD design and manufacturing to the introduction section to strengthen the motivation of our work.
> > >
> > > For designers, our method can save time by addressing the issue of continuously reasoning drawings and verifying parameters during the design process. For example, designers can interactively check whether a certain dimension is aligned across the orthographic projection. For manufacturing personnel, they can directly get the required parameters based on raster image drawings without needing to be proficient in reasoning drawings or using CAD software, thus lowering the barrier to entry for the work.
> > >
> > > We once again extend our sincere gratitude for your guidance on the paper and for the time you’ve dedicated. We will carefully incorporate your suggestions into the revised manuscript.

---

> ### Author Response · Authors · 2025-08-05
> **We are writing to kindly check if you have any remaining questions, concerns, or further suggestions.**
>
> We are writing to kindly check if you have any remaining questions, concerns, or further suggestions. If there are any aspects that require additional clarification or discussion, we would be more than happy to engage in further communication to address them promptly.

---

### Decision · Program_Chairs · 2025-09-17

**Decision:**

Accept (poster)

**Comment:**

This paper proposes a method for reasoning about 3D CAD shapes using canonical orthographic projection images from multiple views. The approach combines reinforcement learning with supervised post-tuning to improve instruction following and semantic extraction. Reviewers identified several strengths: (a) novelty of the research direction, (b) introduction of the new TriView2CAD dataset, (c) significantly better results compared to prior work, and (d) real-world out-of-distribution evaluations. They also noted some weaknesses: (a) experiments focus primarily on a specific CAD category (prefabricated bridge piers) with limited diversity (e.g., cylinder-cuboid configurations), (b) lack of evaluation with industrial VLMs, (c) unclear requirements for prompt engineering, and (d) low real-world accuracy. During the rebuttal, the authors presented additional experiments with industrial VLMs, clarified their prompt engineering methodology, and included new experiments with DeepCAD models to demonstrate generalization and better accuracy compared to alternatives.

Ultimately, four reviewers leaned positive, while one reviewer (jq8V) remained negative, arguing that “the initial paper itself should have included the more complicated dataset ... a thorough evaluation of the approach on the expanded CAD dataset was missing from the initial submission.” However, the authors addressed this concern during the reviewer-author discussion, as also noted in their final remarks. The AC does not share the reviewer’s position that the paper should be rejected solely based on the initial version while disregarding the rebuttal and author-reviewer discussion.

The AC recommends acceptance, and strongly urges the authors to incorporate the additional experiments, results, and clarifications from the rebuttal and discussion (i.e., minor revisions) into the final version of the paper!